# COVID-19: Business Innovation Challenges

**Magdalena Gorzelany-Dziadkowiec**

Department of Organisations Development, Cracow University of Economics, Rakowicka 27,
31-510 Cracow, Poland; gorzelam@uek.krakow.pl

**Abstract:** The goal of this paper was to investigate how the COVID-19 pandemic has affected the readiness and ability to innovate in business. The paper's objective and research questions were pursued with a traditional literature review and an original diagnostic survey using an original questionnaire with a respondent data section and close-ended questions. Responses were collected using the CAWI technique. The primary conclusion was that businesses disturbed by the COVID-19 pandemic were more able to innovate in terms of products and management than those that remained unaffected. Regarding theoretical implications, the author proposed a business model for enterprises operating in the COVID-19 environment. The implications of the model are the practical results of the research.

**Keywords:** innovation during COVID-19; business during COVID-19; product innovation; process innovation; management innovation

## 1. Introduction

Not only has the COVID-19 pandemic been a serious threat to public health, but it is of devastating consequence for other burning social problems such as economic security, democracy, or gender equality. Moreover, in a mere few weeks, it kindled a violent global unemployment crisis. Thus, the pandemic is more than just a matter of public health and will gravely affect many other social issues.

Different nations adopted different strategies to respond to the virus and control the epidemic. They can be identified as one of three categories: strict control with unlimited resources, relentless contribution with limited resources, and rough rationality with limited resources [1]. As a pandemic and its development cannot be forecast, the ability to innovate is not only a decisive factor for the competitive capabilities of business but a driving force of the sustainable development of each country. The ability and readiness of the business to innovate are crucial for responding to environmental changes [2]. Hence, it is of utmost importance to investigate factors that impact business innovations. The research to date focuses on securing government R&D funds [3], the working capital of enterprises, and the effect of external and internal corporate governance mechanisms on business innovation [4–6]. It fails to offer the angle of the ability and readiness of the business to innovate. One could perceive it as a gap in the insight into whether the business innovated during the COVID-19 pandemic, how it reacted to lockdowns, whether pandemic restrictions caused business crises, and what actions were taken to help enterprises survive this difficult time. Moreover, note that many international researchers suggest that innovation investments are procyclical. Some evidence points out that enterprises that retained their innovation capabilities had better survival chances and greater profitability [7].

In light of the above, the goal of the paper is to gain a deeper understanding of how the COVID-19 pandemic has affected the readiness and capabilities to innovate (in terms of products, processes, marketing, organisation, and management) in business. Therefore, the paper poses the following research questions:

- Has the COVID-19 pandemic caused a crisis in all industries (sections of the economy)?
- Was the occurrence of a pandemic-related crisis linked to the size of the enterprise?
- Did the introduction of product, process, marketing, organisational, and management innovations depend on the size of the business and industry?
- Did the introduction of innovations depend on whether or not a business went through a crisis?
- How did businesses respond to crises?
- Did they introduce management innovations?

The objective and research questions of the paper were pursued with a traditional literature review in Scopus and Google Scholar databases on literature published from 2018 to 2021 in English. Both qualitative and quantitative papers were taken into consideration. The papers were filtered according to their usefulness for the purpose of the research (keywords: innovation, innovativeness, COVID-19), resulting in 104 items. Seventy of them that were the most relevant to the paper's subject matter were chosen for the literature review. The rejected articles concerned mostly technology innovations in various countries or REV 4.0 innovation. Furthermore, the author conducted an original diagnostic survey using an original questionnaire with a respondent data section and close-ended questions. Responses were collected using the CAWI technique. The goal of the survey was to analyse the readiness and capabilities of enterprises to innovate during the COVID-19 pandemic. At this point, the questionnaire was a tool to collect facts and opinions regarding the research problem. It resulted in insight into enterprises' capabilities to innovate during the coronavirus pandemic.

The research offers crucial implications for policy makers and business people in Poland regarding responses to changes in the surroundings through innovations, both during the pandemic and in the post-pandemic reality.

## 2. Innovation during the COVID-19 Pandemic—A Literature Review

### 2.1. Implications of the COVID-19 Pandemic for Economies, Societies, and Enterprises

According to the World Health Organization, the first people with COVID-19 were identified on 8 December 2019 [8]. The outbreak caused mobility restrictions; on 23 January 2020, Wuhan imposed a lockdown to contain the new coronavirus [1]. The general public policy was to halt economic activities in most countries [9] temporarily. The coronavirus has caused a dramatic and unprecedented social and economic upheaval since patient zero [10]. It continues to affect the health and safety of employees and employment stability [11].

The current pandemic situation is often referred to in the literature using the "theory of black swan" [12,13], demand and supply shocks [14], "white swan theory", or "grey rhino" [15]. According to the black swan theory, sometimes things happen that were considered impossible until they happened. A black swan is a term used in economic sciences that designates an unexpected event not foreseeable by (almost) anybody. Such events often greatly affect the world and hurt the economy and society. In its current meaning, the term gained popularity after a Lebanese-American researcher N. N. Taleb used it in 2007 [15], stating that black swans increasingly define global events and history due to more and more complex societies. Interactions between factors are often neglected, and forecasts are based on existing patterns and models. According to N. N. Taleb [15] and the theory of black swan, the current COVID-19 pandemic is not a typical black swan. He believes the current crisis to be a white swan or grey rhino. It is an event of tremendous consequences but that are predictable and probable. A characteristic feature of a grey rhino is that investors tend to ignore the threat it entails for a long time or underestimate it. E. Mączyńska [12], N. Rowan and J. G. Laffey [16], Guan et al. [17], and G. Reid, N. O'Beirne, and N. Gibson [18] believe otherwise. They would not hesitate to compare the

current pandemic crisis to a typical black swan. They pointed out that this event was unlikely and unexpected, seemingly impossible. It had a massive impact on restrictions and hygiene policies implemented globally, tangibly affecting our current reality when it did happen. COVID-19 has affected healthcare systems, governments, but also enterprises all over the world. The unprecedented business consequences include market and financial shock [8]. Some industries, such as healthcare, suffer from understaffing [16]. Others apparently have devised new redundancy and training strategies [9,19]. Optimum controls have been employed to model business behaviour—in terms of hiring, discharging, and training employees. It was noted that the manager should lay off the least productive personnel first to reduce costs. As inefficient employees leave, profit improves and can be reinvested in expansion and training [19]. The actions taken to control the coronavirus fundamentally impacted food security [20].

Despite some controversies regarding the theory of N. N. Taleb that black swans drive the world, there is no doubt that actions towards antifragility (which has a singular property of allowing us to deal with the unknown, to do things without understanding them—and do them well) are necessary and justified in the event of a flock of black swans. It is particularly important when the swans mutate into new varieties: green for environmental disasters or blue for unexpected events generated by digital technology and artificial intelligence. Therefore, people in charge of organisations need to prepare for the flock by building, reinforcing, and developing antifragility [12]. Many enterprises turned to sustainable production [21,22] because COVID-19 has influenced many collective behaviours and changed consumer choices [23].

### 2.2. Business Reaction to the COVID-19 Crisis

The COVID-19 crisis has exposed key weaknesses in enterprises and supply chains regarding work conditions and contingency readiness. According to the OECD, how enterprises respond to the COVID-19 crisis would permanently affect their balance sheets and productivity during their recovery. Stimulated by the COVID-19 crisis, enterprises find new ways to survive and grow. This is particularly apparent in small and medium enterprises [24] that turned out to be susceptible to tremor and founded their survival strategies on entrepreneurship [25,26] and innovation [27,28]. Perceiving chaos as an opportunity [29], businesses employed creativity, innovation, and entrepreneurial spirit to solve problems and grasp opportunities in a changed environment [30].

In light of the above, one could conclude that managers need to take advantage of opportunities and be more resilient to change. Resilience is a crisis management concept as it helps understand how enterprises adapt to their surroundings [30].

### 2.3. Innovation—A Response to a Crisis

Innovation can significantly contribute to adaptability. Schumpeter [31] defined it as the introduction of a new product, production method, opening a new market, access to a new source of materials, and reorganisation of an industry. Schumpeter's deliberations were continued by Drucker [32], who defined innovation as a specific entrepreneurial tool, an activity that opens new ways of creating wealth from resources. Kotler [33] believes innovation to be a product, service, or idea that is perceived as something new. The idea may be old, but the key is the perception of the person that considers it new.

A very interesting interdisciplinary approach to the notion and nature of innovation was proposed by Baregheh, Rowley, and Sambrook [34], who pointed out that organisations had to innovate in response to evolving customer expectations, lifestyles and changing technologies, markets, and structures. Zahara and Covin [35] suggested that innovation was a source of life, survival, and growth for the business. In their work, Baregheh, Rowley, and Sambrook [34] discussed various types of innovation (new products, processes, services, and organisational solutions), and various forms, interests, and ways of interpreting innovation in various branches. They proposed a universal definition of in-

novation with a diagram of six attributes: stage (creation, generation, implementation, development, adaptation), social (organisation, enterprise, customers, social system, employees, software developers), means (technology, idea, invention, creativity, market), nature (new, improved, changed), type (product, service, process, technique), and objective (success, rivalry). To identify and express the definition, the authors defined innovation as a multistage process whereby organisations transform their products, services, or processes to grow, compete, and differentiate themselves in the market.

The attitude towards innovation has evolved significantly from the classical doctrine [31–33] to the modern day, where much value is assigned to technological innovation (process and product), non-technological innovation (organisational and marketing), and management innovation.

Today's technological innovation discourse needs to appreciate its importance during the COVID-19 pandemic, which caused rapid growth in the demand for necessary medical equipment, medicines, and high-end IT solutions. Javaid, Haleem, and Vaishya et al. [36] emphasised the immense importance of Industry 4.0 technology, which helps control and manage the pandemic. Some authors suggest that the COVID-19 pandemic could lead to a fifth industrial revolution (society 5.0). The role and impact of Industry 4.0 grow as the world progresses through different stages of the pandemic. Every enterprise is disturbed, which is reflected in the shirking global economic activity and shortage of smart production technologies. Significant technological changes were driven by disasters and outbreaks of infectious diseases [37], hence the tremendous technological changes today.

Innovation-friendly attitudes are clearly visible in products, services, quality, production processes, or management methods [38]. They are becoming the primary creative force of each organisation and need to be embedded into the management system and enterprise culture. Identifying management innovation in the literature was the starting point for defining it as a departure from traditional managerial principles, processes, and practices [39,40]. In other words, management innovations are new solutions for processes, operating principles and methods, and managerial structures that significantly change how the organisation reaches its goals [41]. They include new management practices, processes, structures, or techniques to improve effectiveness [42,43]. Management innovations are crucial because they ensure further innovation to facilitate quick and flexible responses to market signals and challenges, leading to the implementation of the strategy [38,44]. Analysis and research by P. Nakagaki, J. Aber, and T. Fetterhoff [45] yielded conclusions that two important obstacles had to be overcome to drive the innovative capabilities of every large business. The first one is to create the eureka moment, which represents the value of innovative activities in bright colours and demonstrates the role of senior management without questioning it; the other is the shift towards innovation culture.

The managers who participated in the CFO Survey 2020—spring edition by Deloitte, a consulting company, appreciated the destructive impact of the coronavirus on business, which would affect income, jobs, and planned investment projects. The coronavirus pandemic, unexpected and yet taking place in most countries virtually simultaneously, is the pivotal point for the frame of reference of company managers. They revised their fears and embarked on new plans [46]. Organic innovation was the response to business problems. One report entitled "Droga do innowacji a COVID-19" [Path to innovation vs. COVID-19, **48**] looks into challenges related to the development of innovation in business, such as securing of external funding**.** In the time of the COVID-19 pandemic, stronger innovation can be a success driver after the crisis. It consisted particularly of frugal innovation, "good enough" affordable products [47] that meet the needs of consumers with limited resources, which helped most small enterprises in Poland hold their heads above water.

Note here that the bottom line for innovation effectiveness is its verification by the market. P.P. Saviotti and A. Pyka [48] indicated that the backbone of effective innovation

is acceptance by demand. They believed the process of innovating alone may be inconsequential for economic growth if innovative products are not bought but instead, poorly accepted by consumers. The lack of demand for new products is a barrier associated with the implementation of innovation and a ball and chain for the entire innovation process. The innovative effort of enterprises and the market behaviour of consumers are significantly linked beyond any doubt. The relationships are investigated on various levels. K. Włodarczyk [49] identified key research areas regarding modern consumer market behaviour associated with introducing innovation to a market: innovative consumption models, diffusion of innovation, the impact of norms, values, beliefs, and personality traits on the reception of innovation, new technologies, innovating consumer behaviour, models of embracing innovation by consumers, and resistance to innovation from consumer groups.

Analyses of innovative consumption models focus on how consumers use innovative products and services and what consumers know about sustainable development in production [50]. Additionally, consumer environmental preferences need to be considered when designing innovations. Sustainable development of production makes manufacturers strive towards a competitive advantage through appreciating public expectations. The increase in respect for the natural environment in business, noticeable for some years, has become the primary development trend in the supply chain, leading to the emergence of green supply chains [51]. One of the founding fathers who introduced the term into the literature is Beamon [52]. The concept of a green supply chain involves a comprehensive outlook on relationships between the natural environment and production optimisation within the supply chain. Green supply chain management takes into account the entire cycle of product design, production, packing, sale, use, and recycling, including storage, transport, and information flow that should conform to environmental standards [53].

### 2.4. Research Hypotheses

The existing entrepreneurship and innovation practices are evolving to adapt production systems to the reality during and after COVID-19 [28]. As shown in the literature review, entrepreneurship and innovation practices should be conceived in terms of product and process innovation and marketing and organisational novelties. Moreover, management innovation (new management practices) is an important and interesting domain. Note that motivation, positive attitude towards entrepreneurship, risk-taking [54], knowledge [55], and relationships fuel creativity and contribute to product innovation [56]. They can also drive new solutions in production and marketing processes, organisational structure, and management. This background inspired the present research on innovation capabilities of enterprises (business innovativeness) during the COVID-19 pandemic. The evolving innovation paradigm facilitated the classification of innovation as Product, Process, Marketing, and Organisational (organisation and management) innovation both in literature and research.

Product innovation is defined as introducing a new product or service or significantly improving their features and applications [57,58]. They concern the product and involve any changes towards its improvement or diversification of product portfolio. Therefore, the following research hypotheses are proposed:

**Hypothesis 1 (H1):** *Product innovations have been introduced in enterprises where the COVID-19 pandemic crisis occurred.*

Process innovation is new process solutions or the implementation of a new or significantly improved production or delivery method. It includes technology, equipment, and software changes. Sadkowska [59] defined process innovations within three areas. Following this approach, process innovations were assigned to an individual functional area of an enterprise where they are used. The other grouping is innovations defined by their goals; the essence of a process innovation is described through the objective. The last

group is process innovations defined by features of the entity that introduces them. Three research hypotheses are proposed here:

**Hypothesis 2 (H2):** *Process innovations have been introduced in enterprises where the COVID-19 pandemic crisis occurred.*

Marketing innovation is defined as the implementation of a new marketing method involving substantial changes in product design/structure, packaging, distribution, promotion, or pricing strategy. Furthermore, marketing innovation involves introducing e-commerce channels and solutions for mobile shopping [60]. The following research hypotheses are proposed for this domain:

**Hypothesis 3 (H3):** *Marketing innovations have been introduced in enterprises where the COVID-19 pandemic crisis occurred.*

Organisational innovation is the introduction of a new organisational method in the operational policies, workplace organisation, or in relationships with the environment. Czekaj [61] proposed a very detailed view of organisational innovation. He defined his classification as the factual scope of management system improvement from the standpoint of applied organisation science. Lichtarski [62] also investigated organisational structure innovation, pointing out certain components indicative of a shift towards organic and innovative structures. However, the author further noted that it was impossible to determine whether organisational structures of today are innovative unambiguously. They should instead be considered in the broad situational and historical context of each organisation.

**Hypothesis 4 (H4):** *Organisational innovations have been introduced in enterprises where the COVID-19 pandemic crisis occurred.*

Analyses of new possibilities of introducing and classifying innovation refer to management increasingly often. The literature offers views that management innovation contributes to value creation [63] and competitive advantage [64–66] more than the product, marketing, process, or strategy innovation. Hamel and Breen [65] argued that management innovation was in high demand and dubbed the modern management paradigm an "ageing technology". They further pointed out that not much had changed in management over the last few decades. The hierarchical system was flattened but still remained. Line employees were more independent and better trained but still had to conform with management decisions. Junior managers still needed the green light of seniors to be promoted. Strategic decisions were still being taken centrally at the top, while any responsibility was dispersed.

**Hypothesis 5 (H5):** *COVID-19 resulted in management innovations.*

Another investigated area was the relationship between new products, processes, marketing, and management solutions and the innovativeness of enterprises. Therefore, the following research hypothesis is proposed.

**Hypothesis 6 (H6):** *New solutions affect the innovativeness of enterprises.*

## 3. Materials and Methods

### 3.1. Population

The original research was a diagnostic survey using an original questionnaire with a respondent data section and close-ended questions. Responses were collected using the CAWI technique. The questionnaire for the research was validated with a pilot survey,

where respondents were asked about their understanding of the questions in the questionnaire and the correctness of their content. The structure of the questionnaire was discussed with experts, and economics and management professors at the Cracow University of Economics. The questionnaire was uploaded to Google Drive, and the link to it was sent to representatives of randomly selected enterprises. The link was sent by e-mail to about 2000 organisations and additionally posted on social media. To improve the reliability of the research, the questionnaire was aimed at economically active people (employees, business owners, managers) who are practitioners and operate in business circles. The response rate was 31%, with 622 completed questionnaires. The survey period was 20 September 2020 to 30 June 2021. The population included random representatives of various industries (NACE sections): C—manufacturing (manufacture of food products, printing, automotive); D—electricity, gas, steam, and air conditioning manufacture and supply (fuel distribution and trade); E - Water supply; sewerage; waste managment and remediation activities; F—construction (works connected to the construction of buildings); G—wholesale and retail trade, excluding sale of motor vehicles (sale of food, beverages, and tobacco, sale of household equipment); H—transportation and storage (transport by road, warehousing and storage—logistics enterprises); I—accommodation and food service activities (accommodation, catering, food services—restaurants, coffeehouses, ice cream parlours); J—information and communication (publishing activities, motion picture production, data processing—hosting); K - Financial and insurance activities; L—real estate activities; M—professional, scientific and technical activities (legal, accounting, bookkeeping, and tax consultancy); N—administrative and support service activities (HR, tour operators, agents, travel agencies); O—public administration (lecturers, public authority officers); P—education (preschools, schools); Q—health care, residential care, and social work (healthcare); R—arts, entertainment and recreation (creative activities related to culture and entertainment, libraries, physical well-being activities, fitness clubs); S—other service activities (personal service activities—hairdressing, beauty treatment). The structure of the survey population by industry (section) and size is shown in Chart 1.

The number of responses in sections D, I, L, N, P, Q, E, K, and R did not exceed ten. They were omitted in general summaries due to their negligible value. The largest numbers of responses came from representatives of section G—wholesale and retail trade with 142 responders, followed by J with 108 respondents, and C and M with over 90 respondents each. About 40 representatives of sections H, F, O, and S each completed the survey. The largest fraction (32% of the population with 199 respondents) was large enterprises, closely followed by micro enterprises (30%). Next came small enterprises with 23%. The least numerous group were medium-sized enterprises (15%). Nearly 74% of the respondents were employees, 14%—managers, and 12%—owners.

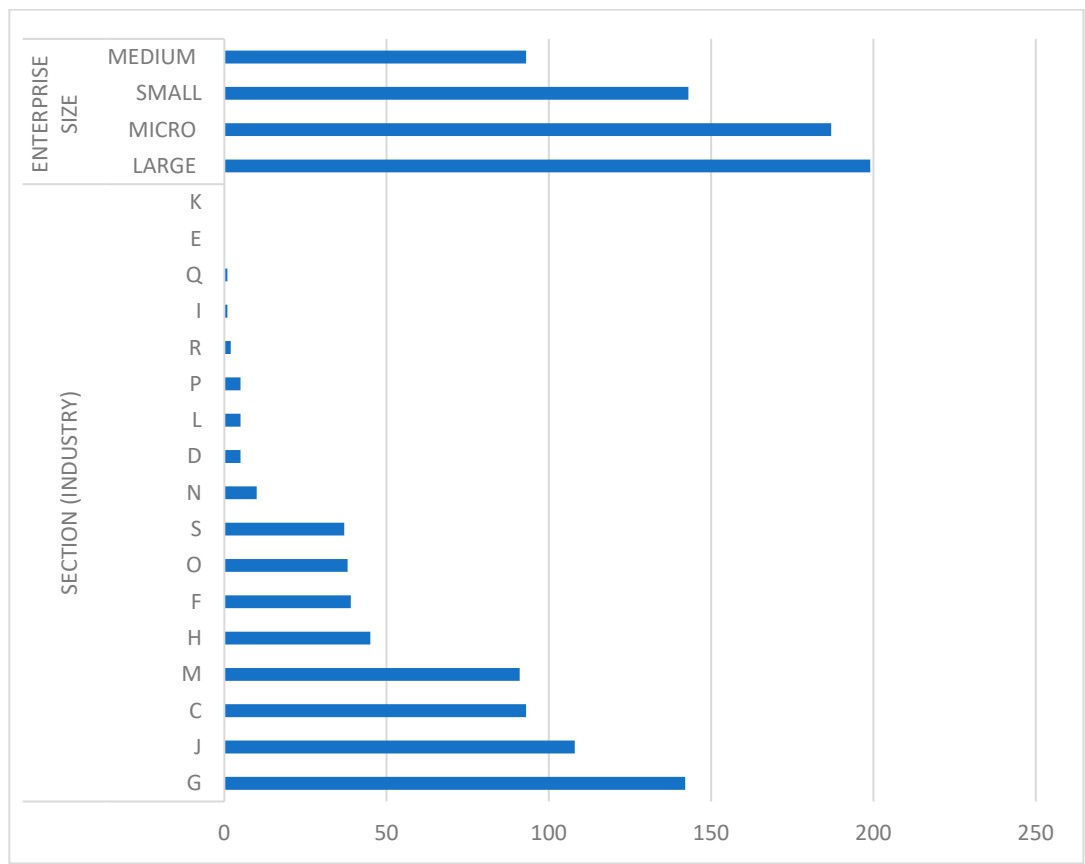

**Chart 1.** The structure of the survey population. Source: original work based on research

*3.2. Measurement*

The measuring tool for the research was an original questionnaire consisting of a respondent data section and a main section. The respondent data section had three questions about the industry (section), size of the enterprise, and role within the organisation (manager, employee, owner). The objective was to identify the respondents so that relations between the size of the enterprise, the role, and core questions could be identified. The main part consisted of seven questions. Two of them were polar questions; a further two offered "yes", "maybe", and "no" as responses. The last three, most complex, questions used a five-point Likert scale [67]. The Likert scale allows researchers to detect even subtle differences in attitudes. The advantage of this scale over simple scales is also apparent in the fact that individual points cannot significantly affect the final result. They are meant to be balanced within the scale, while the specificity of a simple scale can affect research conclusions to a large degree. Normalisation in ranking methods consists of ordering objects according to the ordering criterion for the given variable. Next, variants of the variable are assigned ranks, conventional numeric values that are most often ordinals of positions of the objects in an ordered sequence [68]. Hence, for the present research, 1 means "does not concern" and 5 means "concerns substantially".

The data and relationships were analysed using Spearman's rank correlation coefficient**.** Spearman defined his coefficient as a simple Pearson correlation coefficient for ranks of variables (hence, the name rank correlation coefficient). It describes the strength of correlation of two qualitative and measurable variables in small populations that can be ordered. The measure values lie in the interval of <–1,+1> [69]. The closer it is to one, the stronger the correlation between the variables [70]. The hypotheses were then verified with the chi-squared test for contingency tables (in simple terms, the test is to check whether any statistical differences between response percentages occur). The differences between percentage values in the column are significant when p is equal to or less than

0.05 [71]. Finally, the hypotheses were verified with the Student t-test using three values: t—the statistic; df—the number of degrees of freedom (the sum of questionnaires from enterprises with a crisis and enterprises with no crisis less 2); and p—the probability of the null hypothesis. The null hypothesis always concerns no difference between means. When p is less or equal to the limit value, also referred to as the level of significance (most often 0.05, 0.01, or less), the null hypothesis is rejected, and its alternative is accepted. The null hypothesis was that two juxtaposed means are equal: H0: mean 1 = mean 2. The alternative hypothesis was the opposite: the means differ (or one is greater than the other): HA: mean 1 ≠ mean 2 (or mean 1 > mean 2 or mean 1 < mean 2). The null hypothesis was rejected when the probability p was lower than the assumed significance level $\alpha$. Instead, the alternative hypothesis was accepted as true. Note that the statistical hypotheses are the research hypotheses in formal notation. The alternative hypothesis is the research hypothesis that the paper aims at proving. The null hypothesis is merely an auxiliary hypothesis.

## 4. Results

### 4.1. Analysis and Results

The first step of the analysis was to determine which sections of the economy had been affected by the crisis and whether the introduction of (product, process, marketing, organisational, and management) innovations depended on the size of the enterprise. Correlations between the variables were measured first. The results are summarised in Table 1.

**Table 1.** Pearson's chi-squared and maximum likelihood chi-squared coefficients for the variables.

| Variables | Pearson's Chi-Squared | *p* | Maximum Likelihood Chi-Squared | *p* | df |
|---|---|---|---|---|---|
| Crisis vs. enterprise size | 20.78789 | 0.00012 | 20.96982 | 0.00011 | 3 |
| Crisis vs. section | 29.40782 | 0.00012 | 29.92991 | 0.00010 | 7 |

Note: *p*—the level of significance (likelihood), df—degrees of freedom. Source: original calculations.

The results demonstrate a relationship between the occurrence of the COVID-19 pandemic crisis in an enterprise and its size and section. The level of significance is below 0.05. In light of the above, the next step was to analyse sections and sizes of enterprises hit by the crisis. The results are shown in Table 2.

**Table 2.** Enterprises where COVID-19 caused a crisis.

| | Yes | No |
|---|---|---|
| **Enterprise size** | | |
| Micro (0–9 people) | 62.60% | 37.40% |
| Small (10–49 people) | 49% | 51% |
| Medium (50–249 people) | 38.70% | 61.3 |
| Large (over 250 people) | 42.7 | 57.3 |
| Column total: | 49.50% | 50.50% |
| **Industries (sections)** | | |
| C—manufacturing | 49.46% | 50.54% |
| F—construction | 38.46% | 61.54% |
| G—wholesale and retail trade | 49.30% | 50.70% |
| H—transport and storage | 64.44% | 35.56% |
| J—information and communication services | 60.19% | 39.81% |
| M—professional, scientific, and technical services | 34.07% | 65.93% |
| O—public administration | 28.95% | 71.05% |

| | | |
|---|---|---|
| S—other services | 64.86% | 35.14% |
| Total | 49.52% | 50.48% |

Source: original calculations.

Results in Table 2 demonstrate that the COVID-19 pandemic crisis hit micro enterprises the most (62.6% of the population). Nearly 50% of small enterprises declared that they had been affected by the crisis. The most affected sections were services (64.86% of the population), transport and storage (64.44% of the population), and information and communication (60.19% of the population).

The relationship between the introduction of (product, process, marketing–new packaging, new sale channels, new pricing policy, organisational, or management) innovations and enterprise size is presented in Table 3.

**Table 3.** Enterprise size vs. introduction of innovations (Pearson's chi-squared, maximum likelihood chi-squared).

| | Pearson's Chi-Squared | $p$ | Maximum Likelihood Chi-Squared | $p$ | df |
|---|---|---|---|---|---|
| New products | 22.34753 | 0.03380 | 22.09635 | 0.03645 | 12 |
| Improved products | 39.29739 | 0.00009 | 39.04751 | 0.000010 | 12 |
| New production process solutions | 50.11906 | 0.00000 | 49.59923 | 0.00000 | 12 |
| New packaging | 19.43895 | 0.07847 | 20.45483 | 0.05896 | 12 |
| New sale channels (e.g., online) | 13.72934 | 0.31833 | 13.40207 | 0.34051 | 12 |
| New pricing policy | 18.02392 | 0.11497 | 18.51334 | 0.10097 | 12 |
| Planning | 37.79458 | 0.00017 | 37.82298 | 0.00016 | 12 |
| Organising | 52.83097 | 0.00000 | 52.10566 | 0.00000 | 12 |
| Leadership (motivating, leading) | 52.83097 | 0.00000 | 52.10566 | 0.00000 | 12 |
| Control | 55.23910 | 0.00000 | 54.58167 | 0.00000 | 12 |

Note: $p$—the level of significance (likelihood), df—degrees of freedom. Source: original calculations.

The data in Table 3 demonstrate a relationship between the introduction of product, process, organisational, and management innovations and the size of the enterprise (significance level below 0.05). Furthermore, the analysis shows that management innovations (planning, organising, leadership–motivating and leading, and controlling) depend on the enterprise size. Therefore, the next step was to investigate which innovations were introduced in micro, small, medium, and large enterprises. The results are presented in Chart 2.

Data in Chart 2 show that new and improved products were introduced most often in large enterprises (almost 50% of the answers were "yes" and "to a large degree"). On the other hand, new production process solutions were implemented mostly in micro enterprises (about 40% of them). Nevertheless, the other types did not innovate in this area.

Chart 3 shows enterprises' abilities to introduce management innovations regarding planning, organisation, leadership, and control.

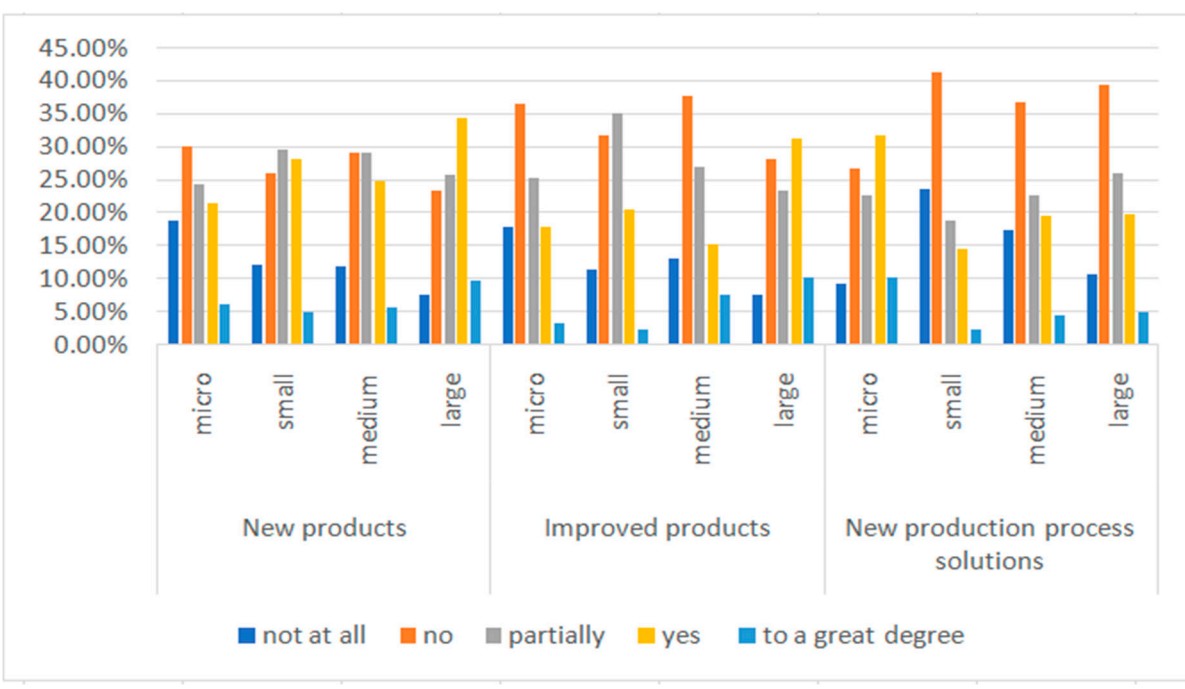

**Chart 2.** Product and process innovation vs. enterprise size. Source: original work based on research.

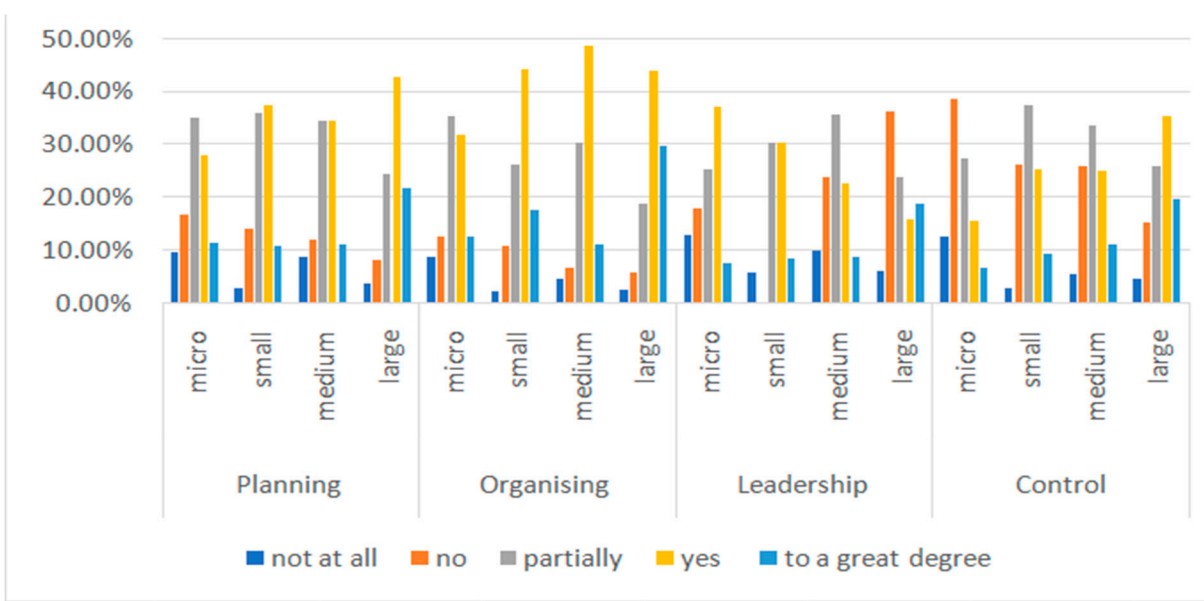

**Chart 3.** Management innovation vs. enterprise size. Source: original work based on research.

New planning solutions were most apparent in large enterprises (about 65% of respondents) and implemented in small and medium enterprises (nearly 50% of respondents indicated such changes). Organisational structure changes were the most evident in large enterprises (over 40% of the respondents declared "yes" and 30% "to a large degree"). Structural changes were apparent in small enterprises (45% "yes" and almost 20% "to a large degree") and medium enterprises (almost 50% "yes" and 10% "to a large degree"). Micro and small enterprises declared changes in leadership (motivation and leading) (about 50% "yes" and "to a large degree"). In large enterprises, the respondents declared no changes in this area (36% "no" and 6% "not at all"). New control solutions were most apparent in large enterprises (almost 70% of the respondents indicated some changes) and, to some extent, in small and medium enterprises. Micro enterprises did not introduce changes in control (over 50% declared "no" and "not at all").

The results of the correlations between the industry (section) and innovation are shown in Table 4.

**Table 4.** Section vs. introduction of innovations (Pearson's chi-squared, maximum likelihood chi-squared).

| | Pearson's Chi-Squared | *p* | Maximum Likelihood Chi-Squared | *p* | df |
|---|---|---|---|---|---|
| New products | 44.57253 | 0.02436 | 49.40257 | 0.00753 | 28 |
| Improved products | 40.59601 | 0.05846 | 46.00449 | 0.01741 | 28 |
| New process solutions | 31.69638 | 0.28704 | 36.44088 | 0.13171 | 28 |
| New packaging | 62.18367 | 0.00021 | 70.63944 | 0.00002 | 28 |
| New sale channels (e.g., online) | 54.46790 | 0.00197 | 60.51265 | 0.00035 | 28 |
| New pricing policy | 56.05525 | 0.00127 | 61.77072 | 0.00024 | 28 |
| Planning | 32.50338 | 0.25451 | 32.94326 | 0.23784 | 28 |
| Organising | 24.47934 | 0.65600 | 25.47430 | 0.60192 | 28 |
| Leadership (motivating, leading) | 33.66772 | 0.21203 | 34.25059 | 0.19274 | 28 |
| Control | 36.65005 | 0.12678 | 35.92897 | 0.14442 | 28 |

Note: *p*—the level of significance (likelihood), df—degrees of freedom. Source: original calculations.

The summary in Table 4 suggests that the implementation of product innovations (new products) and marketing innovations (new packaging, new sale channels, new pricing policy) depends on the industry (section), significance level below 0.05. On the other hand, management innovation (planning, organisation, leadership, and control) and process innovation do not depend on the section, significance level above 0.05. A detailed analysis of innovations in correlated sections is shown in Chart 4.

The summary in Chart 4 shows a connection between product and marketing innovations (sale channel, new packaging, pricing policy) and the industry (section). Still, new products were launched only in S (services), G (wholesale and retail trade), and C (manufacturing). No section introduced new packaging. All respondents answered "no" or "not at all". The same applies to new sale channels. New pricing policies were impmented in F (construction) and J (information and communication).

### 4.2. Verification of the Hypotheses

The research focused on investigating how the COVID-19 pandemic has affected business's readiness and ability to innovate. Results were described by recoding responses into natural numbers: to a large degree—5; yes—4; partially—3; no—2; not at all—1. The same coding was applied to questions with only three answers (no, partially, yes): no—2; partially—3; yes—4. The neutral answer is 3. Answers below 3 are negative (worse, less, to a lesser extent), and answers above 3 are affirmative (better, more, to a larger extent). The results are shown in Table 5.

**Table 5.** Hypotheses and results for enterprises affected by the crisis.

| | Means Tested against a Reference Value. Threshold Condition v9 = 'yes'. | | | | | | | |
|---|---|---|---|---|---|---|---|---|
| | Mean | SD | Valid | SE | Reference | t | df | *p* |
| H: 1 | 2.8 | 1.15 | 308 | 0.07 | 3.00 | −2.270 | 307 | 0.0239 |
| H: 2 | 2.7 | 1.11 | 308 | 0.06 | 3.00 | −5.062 | 307 | 0.0000 |
| H: 3 | 3.2 | 0.81 | 308 | 0.05 | 3.00 | 3.916 | 307 | 0.0001 |
| H: 4 | 3.2 | 1.26 | 308 | 0.07 | 3.00 | 3.300 | 307 | 0.0011 |
| H: 5 | 3.1 | 1.21 | 308 | 0.07 | 3.00 | 1.985 | 307 | 0.0480 |

| H: 6 | 3.3 | 0.76 | 308 | 0.04 | 3.00 | 6.503 | 307 | 0.0000 |

Note: *p*—the level of significance (likelihood), df—degrees of freedom. Source: original calculations.

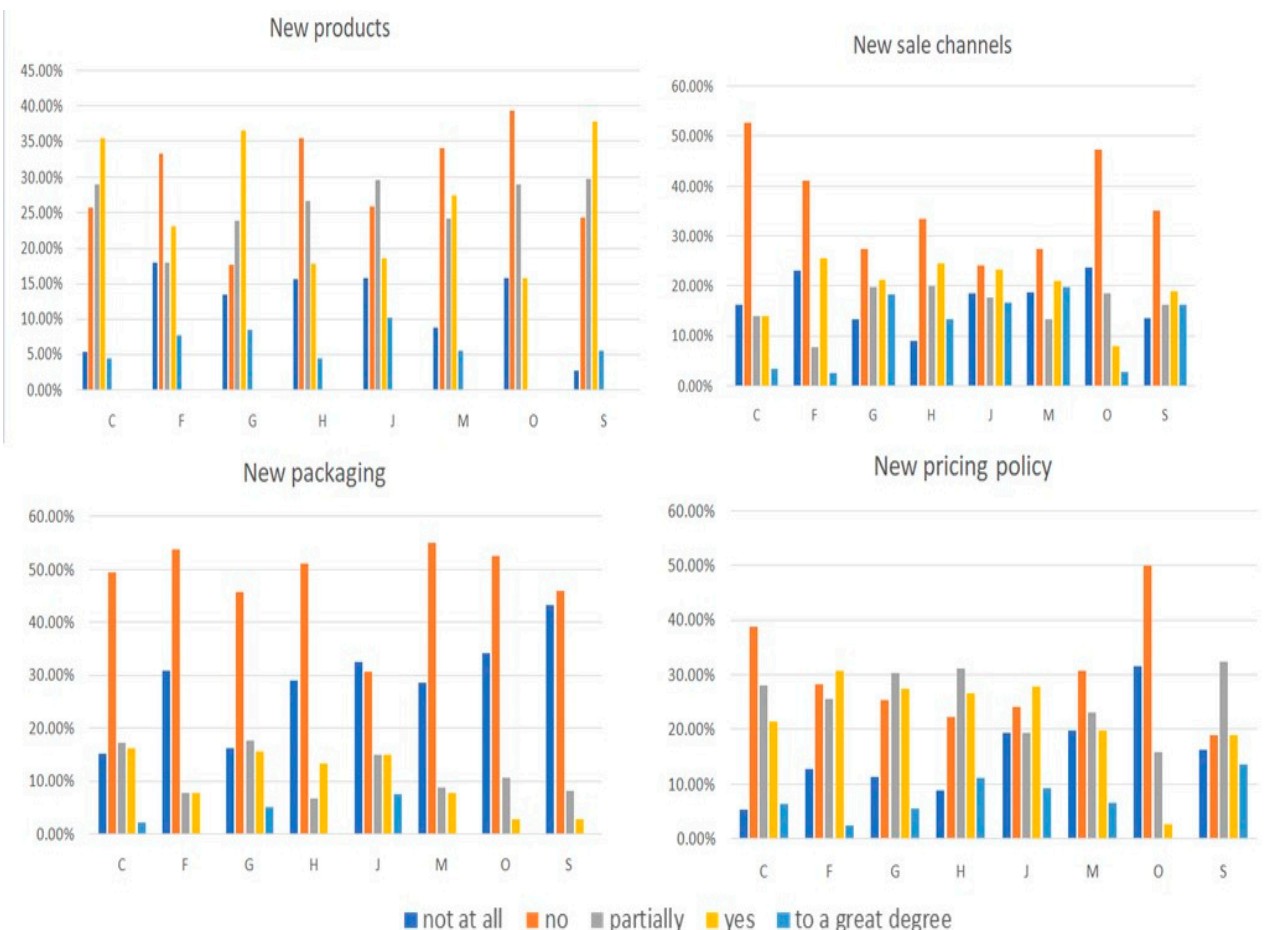

**Chart 4.** The structure of the survey population. Source: original work based on research.

The mean was compared to 3 as if nothing changed during the COVID-19 pandemic. An important concluding factor was the probability p. A difference between the time before COVID-19 and during the pandemic can be identified if p is equal to or less than 0.05. Additionally, the mean has a confidence interval that can be considered to contain the actual mean, not the one estimated from the survey data with a 95% probability. When the confidence intervals of means partially overlap, the means are statistically identical. If the confidence intervals are disjoint, the means differ. The results are shown in Chart 5 to visualise the analyses.

The analyses indicate that all the hypotheses have been confirmed. This means that product, process, organisational, marketing, and management innovations have been introduced in enterprises where the COVID-19 pandemic crisis occurred and new solutions affect business innovativeness.

Additionally, the author verified whether enterprises that did not suffer from the COVID-19 crisis introduced product, process, organisational, marketing, or management innovations by testing the means against the constant reference value (condition v9 = "no"). This analysis demonstrated that enterprises, where the COVID-19 pandemic crisis did not occur, did not introduce product or management innovations.

As the research identified relationships between the introduction of innovations and occurrence of a crisis, a comparative analysis of the means for enterprises with the

COVID-19 pandemic crises and without them was conducted. The results are shown in Table 6.

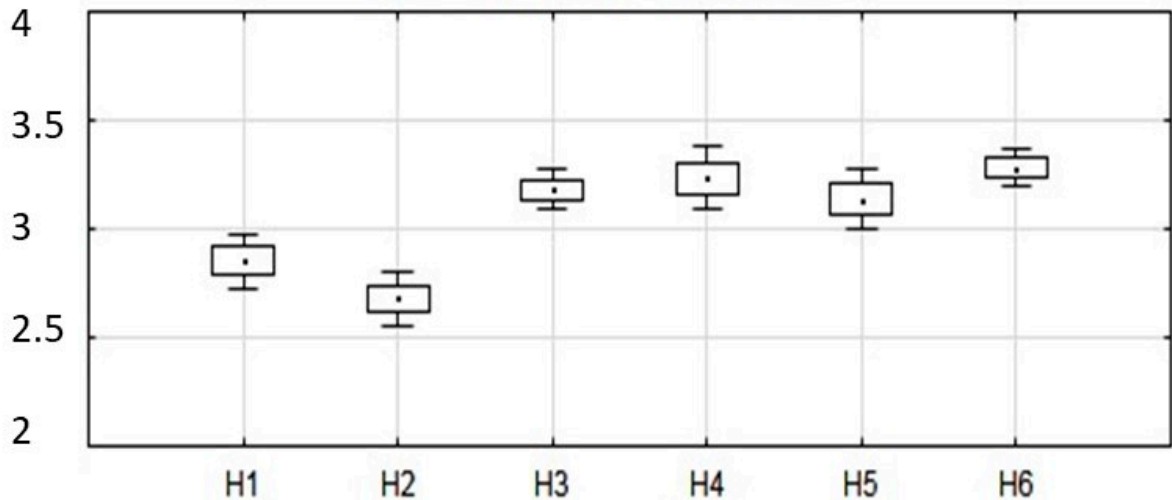

**Chart 5.** A box plot of means and confidence intervals for enterprises affected by the COVID-19 crisis. Source: original work based on research.

The summary in Table 6 shows that enterprises affected by the crisis had a greater ability to innovate than those unaffected. The differences included marketing innovations (packaging, new sale channels, pricing policy), but the mean for these changes did not exceed 3, which means they were not implemented to a large extent. Management innovations (planning, organisation, leadership, and control) were more popular than others, but this mean was not greater than 4 in any area, which means they were not introduced to a large extent. Note that the significance level for planning, organising, and leadership was higher than 0.05, which means there were no significant differences in these areas during the COVID-19 pandemic and before it. The only differences were identified for control and social media use for marketing purposes. The analysis shows that the most apparent changes occurred in organisation (mean 3.63) and planning (mean 3.48).

**Table 6.** Comparative analysis of means.

| Variable | Group 1: no (Crisis-Free); Group 2 Yes (Affected by Crisis) | | | | |
|---|---|---|---|---|---|
| | Mean for Group 1 no | Mean for Group 2 yes | $t$ | df | $p$ |
| New products | 2.93 | 2.85 | 0.900 | 620 | 0.3682 |
| Improved products | 2.81 | 2.72 | 1.069 | 620 | 0.2855 |
| New process solutions | 2.68 | 2.68 | −0.002 | 620 | 0.9980 |
| New packaging | 2.07 | 2.28 | −2.542 | 620 | 0.0113 |
| New sale channels (e.g., online) | 2.69 | 2.92 | −2.187 | 620 | 0.0291 |
| New pricing policy | 2.55 | 3.01 | −5.013 | 620 | 0.0000 |
| Organisational structure changes | 2.71 | 3.24 | −5.429 | 620 | 0.0000 |
| Planning | 3.32 | 3.48 | −1.824 | 620 | 0.0686 |
| Organising | 3.58 | 3.63 | −0.569 | 620 | 0.5697 |
| Leadership (motivating, leading) | 2.98 | 3.14 | −1.688 | 620 | 0.0920 |
| Control | 2.98 | 3.22 | −2.727 | 620 | 0.0066 |
| Social media | 2.90 | 3.18 | −4.148 | 620 | 0.0000 |
| Significant remote work | 3.07 | 2.94 | 1.839 | 620 | 0.0663 |
| New messengers and communication platforms | 3.19 | 3.08 | 1.512 | 620 | 0.1311 |

| New solutions vs. enterprise's innovativeness | 3.39 | 3.28 | 1.852 | 620 | 0.0646 |

Source: original work based on research.

## 5. Discussion

The research and analyses helped answer the research questions and verify the research hypotheses. Regarding the first and second research questions, the COVID-19 pandemic did not cause crises in all the sections. Moreover, the businesses affected by the COVID-19 crisis the most were micro and small enterprises. This finding is consistent with research by J. Męcin and P. Potocki [72], who demonstrated that the smaller the enterprise, the worse it suffered from the COVID-19 pandemic. The sections most affected by the COVID-19 pandemic were services, transport and storage (logistics), and information and communication services (publishers, moving picture production). The present research confirmed conclusions by M. Dzierżanowski [73] that not all industries have been affected by the crisis and by H. Gehrke-Gut [74,75] that globally, the most affected sections are services, publishing, and moving picture production. When forced to pause production, close points of sale, or reduce staff, businesses first look for savings by putting on ice tasks that the managers believe to be non-crucial [76].

Regarding the third research question, the implementation of product, process, and management innovations depended on the size of the enterprise, while marketing innovation was independent of it. This is due to the nature of small, medium, and large enterprises.

Regarding the fifth research question, enterprises that were affected by the crisis did implement new management solutions.

The comparative analysis of the means yielded interesting results. Enterprises affected by the COVID-19 pandemic crisis implemented more innovative solutions than those that steered clear of the crisis. The most significant changes were noted in structures, mostly regarding personnel. According to statistical data, only 31% of the enterprises did not plan to reduce personnel and did not do it. The remaining 69% planned 20–30% reductions. Most lay-offs took place in medium and large enterprises. Therefore, one can expect structural changes to be the most evident in the present research. Next, new solutions were introduced regarding planning (mean 3.48), control (mean 3.22), leadership (mean 3.1) and use of social media for marketing. On the one hand, management innovations are evident: the mean is greater than 3, meaning better, more. On the other hand, the mean never reaches 4, which would mean a satisfactory level of innovation.

The research supported the conclusion that enterprises capable of innovation would have better ways of handling uncertainty during the COVID-19 pandemic. Therefore, the business should improve its innovation capabilities. The result is consistent with research by experts at the Polish Agency for Enterprise Development [77], who analysed actions taken globally by various countries to prevent the consequences of the COVID-19 pandemic crisis. According to their report, an innovative business approach to the unstable environment was a must during the pandemic. Experts also confirmed the growing importance of modern technologies, digitalisation, and sustainable development efforts [77] (orientation towards ecological preferences of consumers and green supply chains). Experts at the general meeting of the World Technopolis Association [78] reached similar conclusions and pointed to innovation as the best way of combating the crisis. Babina, Bernstein, and Mezzanotti [79] noted that financial crises could act both as destructive and creative forces for innovation and provided the first systematic evidence of the role of anxiety in the long-term organisation of innovative businesses. Moreover, the ability to innovate is one of the key features of competitive, dynamic, and progressive organisations [80].

Furthermore, the analyses suggest that the introduction of innovation is the responsibility of managers, leaders oriented towards people and change. Hameed, Nisar, and Wu [81] discussed the link between leaders and innovations and suggested that leaders should be oriented towards knowledge. The authors believed that leadership is among

the most potent sources of increased organisational effectiveness by developing knowledge infrastructure, leading to the strengthening of innovative solutions. Over recent years, researchers have investigated how management practices and systems facilitated innovativeness. Some demonstrated that knowledge management is an important backbone of business innovation [80–84]. Therefore, leadership-oriented managers will look for knowledge internally and externally to apply it to new and much-improved products, processes, organisational structures, and marketing and management.

The present research confirmed that enterprises responded organically and introduced new solutions, but the responses cannot be considered sufficient. The outbreak surprised even the finest strategists. One could hardly expect a clear assessment of the threat, proposals of on-point contingency scenarios, or bold and often painful decisions. The pandemic promoted business environment variability, uncertainty, incomprehension of new problems, complexity (including chaos and confusion, information flood), and the necessity to tell information noise from important facts to the rank of a new business environment. Times of uncertainty need supportive leaders to act. The dictator leader is even more dangerous in volatile times than usually. They create anxiety, decision paralysis, and stall the company when it should be steered like a sailing boat. Supportive leaders kindle trust and provide room for experiments and mistakes, so that crisis strategies can be developed and implemented faster. They help with joint effort and reinforce values that are the backbone for future transformation. It is time for leaders who are ready to accept that the new uncertainty is the only certain thing and can convince employees that there is potential for success in these conditions. The research showed that leadership (leading, motivation) changes were introduced, if only to a limited extent.

## 6. Conclusions

Boards and owners of enterprises now face the responsibility of ensuring liquidity and preserving jobs. It may be the first time some of them came across such a substantial uncertainty regarding the future. They will have to make strategic-level decisions that will determine whether and how fast their business will recover from the crisis. Their success will hinge mostly on the flexible adaptation of the enterprise to market changes, an achievement for which innovation may be the key [85]. The pandemic can only kindle innovation: organisations do not grow weaker searching for innovation to boost effectiveness and optimisation, often technology-based, but embark on the search for business model innovations with new energy to grow agile and resilient to the competitors [85,86], but most of all, to survive. The analyses demonstrated that innovation is the potion of survival [24–28,30,34], so modern business models should be based on them. Again, product, process, organisational, marketing [31–33] and management [34–36,87] innovations should be the primary focus.

The COVID-19 pandemic has changed the lives of people and the functioning of businesses worldwide. Unfortunately, managers were forced to take quick, and more than once wrong, decisions. As was already mentioned, innovation is a key process within the organisation. Without new products, modification of production processes, changes in organisation, marketing, and management, the organisation cannot survive regardless of its functional profile [88]. The present research facilitated a business model for business managers for the time of the COVID-19 pandemic. It is shown in Figure 1 (diagram).

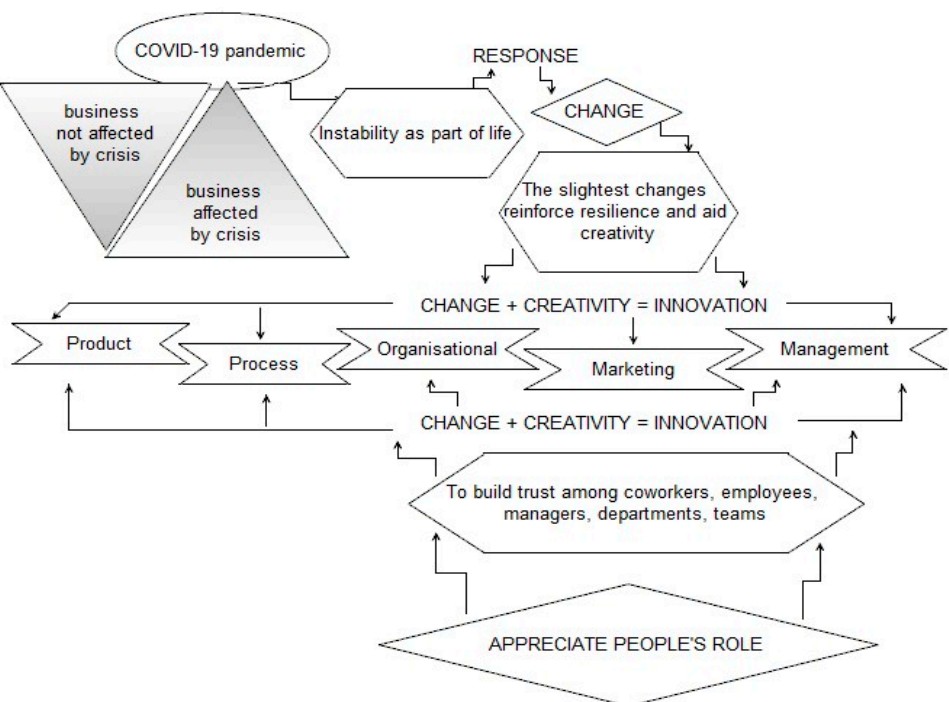

**Figure 1.** The business model for enterprises during the COVID-19 pandemic. Source: original work based on the literature review and original research.

The model in Figure 1 suggests that the COVID-19 pandemic causes crises in some enterprises while not in others. It has also destabilised financial markets, public life, and business operations. Instability should now be considered part of our life. Therefore, both people and enterprises should learn how to live under such conditions, which is through change. Changes should be introduced in small steps at a time as the minimal change reinforces resilience and creativity. Combined with creativity, a change will trigger the readiness and ability to implement product, process, marketing, organisational, and management innovations. According to M. Annunziata and H. Bourgeois [89], we fail to appreciate the role of people in virtually any economic context. From construction to production businesses, we tend to be in awe of technology and take the impact of people on productivity for granted. Moreover—or because of it—we do not see the importance of long-term investment in talent. M. Annunziata believed that underinvested human resources bring poor consequences: a progressing stratification of competencies and incorrect growth of human capital. Therefore, leaders should adopt a different approach to innovation in their business models by appreciating roles of people in organisations, building trust to fuel change and creativity, and improve the readiness and ability to innovate.

It is not enough to determine the business model. Its implementation is also a critical stage. As shown in the business model, the ability to innovate is primarily based on the ability to change combined with creativity. Thus, the four leading components for the business model are leaders, the ability to change, creativity, and innovation [90–93]. The present research facilitated a business model for which the steps to implement it are illustrated in Figure 2.

The model in Figure 2 shows consecutive or simultaneous steps recommended for managers to introduce changes and new business strategies. The action should start with determining people-oriented leaders who appreciate the human potential. The management regime should evolve from autocrats towards leaders who lead—inspire people, initiate changes, learn from mistakes, and favour teamwork. They should embrace creativity [29] with its unobstructed flow of information [82–84], acceptance of risk and risk-taking,

and recognition and rewarding of the slightest new solutions. Changes in behaviour, attitude towards the employee, and triggering changeability and creativity will be the beginning of changes, of innovation.

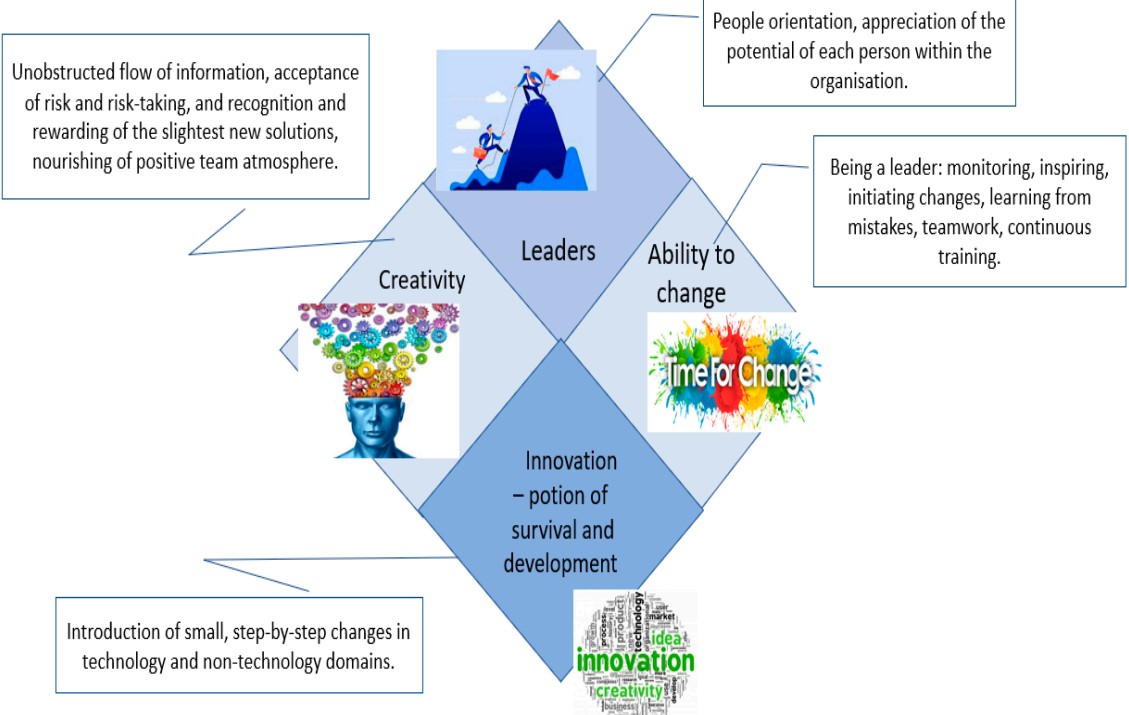

**Figure 2.** Business model implications. Source: original work based on the literature review and original research.

The limitation of the research is that it investigated a group of random enterprises represented by agents of selected industries. The results confirmed findings in other papers and statistical reports but did not facilitate general conclusions. However, they provide a basis for in-depth research concerning specific sectors that would analyse business strategies and their impact on business competitiveness and innovativeness during the COVID-19 pandemic. Future research could also attempt to tackle the question of how much leadership affects the readiness and ability to innovate and what competencies are desirable in managers responsible for innovation.

**Funding:** These studies were financed from the own funds of the Cracow University of Economics.

**Institutional Review Board Statement:** Not applicable.

**Informed Consent Statement:** Informed consent was obtained from all subjects involved in the study.

**Data Availability Statement:** The datasets generated and analyzed during the current study are not publicly available due to participant confidentiality but are available from the corresponding author on reasonable request.

**Conflicts of Interest:** The authors declare no conflict of interest.

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
