# Peer review of "COVID-19: Business Innovation Challenges"

_sustainability, doi:10.3390/su132011439_

Round 1

Reviewer 1 Report

  1. Through the questionnaire survey, 2000 companies were randomly selected to conduct the survey by email, but only 622 questionnaire responses were received. A single survey method results in less data received. Several other survey methods should be added. Collecting data through multiple channels can obtain more comprehensive data.
  2. The data source of this article is the questionnaire, and then analyze the conclusion based on the obtained data. Therefore, the authenticity of the data is extremely important and is the prerequisite for the analysis to reach the conclusion. However, whether the data obtained through the questionnaire is true or not? In order to ensure its authenticity, the author should add an explanation.
  3. The related works about product innovation and managment innovation shoule be further reviewed, such as A combined fuzzy DEMATEL and TOPSIS approach for estimating participants in knowledge-intensive crowdsourcing; Measuring knowledge diffusion efficiency in R&D networks; Selection of manufacturing enterprise innovation design project based on consumer’s green preferences; Decision-making and coordination of green closed-loop supply chain with fairness concern
  4. The author believes that during the COVID-19 pandemic, innovative companies will have better ways to deal with uncertainty, and therefore companies should enhance their innovation capabilities. This result is based on the collected questionnaires. In order to enhance the credibility of its conclusions, some examples should be added for comparison, which is clearer and more intuitive.
  5. The author proposes and introduces a variety of innovative methods, including business model innovation, management innovation, etc. However, it is not enough to introduce the priority of various methods, and you can introduce it in detail in this aspect.

Based on the above comments, it is recommended to review carefully after careful revision.

Author Response

RECENZENT 1

Dear Professor,

Thank you very much for all the comments and recommendations in the review. They are invaluable and helped improve the scientific value of the paper. Allow me to clarify certain issues:

  1. Through the questionnaire survey, 2000 companies were randomly selected to conduct the survey by email, but only 622 questionnaire responses were received. A single survey method results in less data received. Several other survey methods should be added. Collecting data through multiple channels can obtain more comprehensive data.

A more detailed description of who was sent the link to the questionnaire has been added (line 335)

  1. The data source of this article is the questionnaire, and then analyze the conclusion based on the obtained data. Therefore, the authenticity of the data is extremely important and is the prerequisite for the analysis to reach the conclusion. However, whether the data obtained through the questionnaire is true or not? In order to ensure its authenticity, the author should add an explanation.

A clarification has been added that the questionnaire was proposed to economically active people (employees, owners of businesses, managers) – line 335-336

  1. The related works about product innovation and managment innovation shoule be further reviewed, such as A combined fuzzy DEMATEL and TOPSIS approach for estimating participants in knowledge-intensive crowdsourcing; Measuring knowledge diffusion efficiency in R&D networks; Selection of manufacturing enterprise innovation design project based on consumer’s green preferences; Decision-making and coordination of green closed-loop supply chain with fairness concern

The content and literature review have been expanded, lines 206–233 and 647–657.

  1. The author believes that during the COVID-19 pandemic, innovative companies will have better ways to deal with uncertainty, and therefore companies should enhance their innovation capabilities. This result is based on the collected questionnaires. In order to enhance the credibility of its conclusions, some examples should be added for comparison, which is clearer and more intuitive.

Lines 631-647 have been expanded.

  1. The author proposes and introduces a variety of innovative methods, including business model innovation, management innovation, etc. However, it is not enough to introduce the priority of various methods, and you can introduce it in detail in this aspect.

Lines 666-682, 684-687, and 713-717 have been expanded.

Reviewer 2 Report

This is a well written and interesting paper focusing on an area where there is need for more knowledge. The paper is very well structured with significant contribution to current body of knowledge. Research subject and problem have been defined correctly. The content of the article corresponds to the main goal - substantive compatibility in theoretical, methodological and empirical terms. The results of the analysis are described correctly.

The following suggestions can be considered: 

The aim of the paper is said to be: “to provide a more profound understanding of how the COVID-19 pandemic has affected the readiness and ability to innovate in business”/”to gain a deeper understanding of how the COVID-19 pandemic has affected the readiness and ability to innovate in business”. I suggest to use the phrase “to explore how the COVID-19 pandemic has affected the readiness and ability to innovate in business” In my opinion it will be reflect better the empirical process and findings presented in the paper.

In section 2.3. regarding innovations , In order to explain the idea and the nature of innovation I suggest to read and cite the paper of Baregheh, Rowley & Sambrook, 2009 (Baregheh, A., Rowley, J. & Sambrook, S.,(2009) Towards a multidisciplinary definition of innovation. Management Decision. 47 (8): 1323-1339) who have conducted a comprehensive analysis of several innovation definitions (extracted from a number of different disciplines).

In the lines 51-52 there is a grammatical error in the sentence. There is: “Did the introduction of product, process, marketing, organisational, and management innovations depended on the size of the business and industry?” and should be: “Did the introduction of product, process, marketing, organisational, and management innovations depend on the size of the business and industry?”.

In regard to the hypotheses formulated in the article: I think that there are too many hypotheses (15). The statements used in some of the hypotheses (e.g. H8, H9, H11, H12, H13 or H14) seem very simple and obvious, like: “COVID-19 has led to new organisational solutions”, “COVID-19 has led to new control solutions”, “Social media is used for marketing purposes more than before in enterprises where the COVID-19 pandemic crisis occurred” or “There is more remote work than before in enterprises during the 252 COVID-19 pandemic”. It seems obvious that during the pandemic time new organisational and control solutions have occurred. Similarly, of course there was more remote work in almost all organizations and more social media used for marketing”. Summing up, I feel that some hypotheses are not necessary in the paper.

While presenting Materials and Method as well as Measurement sections, the is no information provided on the validation of the questionnaire used for the research. How was it validated? Was there any pilot study conducted etc.? Please provide some information about it.

Discussion section has been combined with Conclusions. Please mark out Discussion section as a separate paper’s chapter. The Discussion part of the paper is too short. This section is crucial for academic papers and should be about contrasting the manuscript's findings with the existing literature. I understand that there are scarce research on the COVID-19 impact on firms’ ability and readiness to innovate but please try to supplement Discussion section with the information to what extent your results are aligned with the evidence found in the adequate literature. To what extent your results are not aligned. Please provide a more comprehensive discussion on this.

Author Response

RECENZENT 2

Dear Professor,

Thank you very much for all the comments and recommendations in the review. They are invaluable and helped improve the scientific value of the paper. Allow me to clarify certain issues:

This is a well written and interesting paper focusing on an area where there is need for more knowledge. The paper is very well structured with significant contribution to current body of knowledge. Research subject and problem have been defined correctly. The content of the article corresponds to the main goal - substantive compatibility in theoretical, methodological and empirical terms. The results of the analysis are described correctly.

The following suggestions can be considered: 

The aim of the paper is said to be: “to provide a more profound understanding of how the COVID-19 pandemic has affected the readiness and ability to innovate in business”/”to gain a deeper understanding of how the COVID-19 pandemic has affected the readiness and ability to innovate in business”. I suggest to use the phrase “to explore how the COVID-19 pandemic has affected the readiness and ability to innovate in business” In my opinion it will be reflect better the empirical process and findings presented in the paper.

The objective has been revised.

In section 2.3. regarding innovations , In order to explain the idea and the nature of innovation I suggest to read and cite the paper of Baregheh, Rowley & Sambrook, 2009 (Baregheh, A., Rowley, J. & Sambrook, S.,(2009) Towards a multidisciplinary definition of innovation. Management Decision. 47 (8): 1323-1339) who have conducted a comprehensive analysis of several innovation definitions (extracted from a number of different disciplines).

The literature has been expanded, including with the suggested paper, lines 147–161.

In the lines 51-52 there is a grammatical error in the sentence. There is: “Did the introduction of product, process, marketing, organisational, and management innovations depended on the size of the business and industry?” and should be: “Did the introduction of product, process, marketing, organisational, and management innovations depend on the size of the business and industry?”.

The phrase has been corrected.

In regard to the hypotheses formulated in the article: I think that there are too many hypotheses (15). The statements used in some of the hypotheses (e.g. H8, H9, H11, H12, H13 or H14) seem very simple and obvious, like: “COVID-19 has led to new organisational solutions”, “COVID-19 has led to new control solutions”, “Social media is used for marketing purposes more than before in enterprises where the COVID-19 pandemic crisis occurred” or “There is more remote work than before in enterprises during the 252 COVID-19 pandemic”. It seems obvious that during the pandemic time new organisational and control solutions have occurred. Similarly, of course there was more remote work in almost all organizations and more social media used for marketing”. Summing up, I feel that some hypotheses are not necessary in the paper.

The hypotheses have been revised and reduced from 15 to 6. Lines 254-332.

While presenting Materials and Method as well as Measurement sections, the is no information provided on the validation of the questionnaire used for the research. How was it validated? Was there any pilot study conducted etc.? Please provide some information about it.

The section has been expanded with questionnaire validation. Lines 329-333.

Discussion section has been combined with Conclusions. Please mark out Discussion section as a separate paper’s chapter. The Discussion part of the paper is too short. This section is crucial for academic papers and should be about contrasting the manuscript's findings with the existing literature. I understand that there are scarce research on the COVID-19 impact on firms’ ability and readiness to innovate but please try to supplement Discussion section with the information to what extent your results are aligned with the evidence found in the adequate literature. To what extent your results are not aligned. Please provide a more comprehensive discussion on this.

The Discussion section has been isolated. Descriptions of study findings have been added and the results have been confronted with the literature. Lines 631-657.

Reviewer 3 Report

It is unclear why publications with technological innovations and REV 4.0 innovations have been excluded when process and product innovations are directed in the spirit of these trends. The authors discuss a search of 70 professional works; the article lacks an overview of research focused primarily on innovation activities.

In Chapter 2.4, I recommend emphasizing the division of hypotheses according to the type of innovation.

In Chapter "2.1 Consequences of the COVID-19 pandemic on economies, societies and businesses" is not a logical sentence: The actions taken to control the coronavirus fundamentally impacted 109 food security and food access for impoverished and marginalized households and communities [21].

Explain more precisely statistical methodology, such as accepting and rejecting null hypotheses.

Table no. 2 is unreadable.

In the text informing about the composition of the industry, it would be appropriate to state their full name instead of the abbreviations of the industry. Chapter title 3. "Population", does not reflect the actual contents of the chapter.

The sentence " The unprecedented business consequences include market and financial shock [8], understaffing, and disturbed supply chains [16].“. There indicates that staff shortages were noted during the pandemic. Would you please specify which sector has experienced a need of employees, or the reasons for their lack? According to the development, it is impossible to consider this state of affairs as a dogma; there are known cases where industries such as hotels, restaurants, spas, air services have started redundancies. The car industry has introduced shortened operating modes.

Author Response

Dear Professor,

Thank you very much for all the comments and recommendations in the review. They are invaluable and helped improve the scientific value of the paper. Allow me to clarify certain issues:

It is unclear why publications with technological innovations and REV 4.0 innovations have been excluded when process and product innovations are directed in the spirit of these trends. The authors discuss a search of 70 professional works; the article lacks an overview of research focused primarily on innovation activities.

The manuscript now includes a more detailed description of technological innovation relevant to REV 4.0, lines 166–177.

In Chapter 2.4, I recommend emphasizing the division of hypotheses according to the type of innovation.

Hypotheses have been grouped by type; lines 254–322.

In Chapter "2.1 Consequences of the COVID-19 pandemic on economies, societies and businesses" is not a logical sentence: The actions taken to control the coronavirus fundamentally impacted 109 food security and food access for impoverished and marginalized households and communities [21].

The sentence has been removed

Explain more precisely statistical methodology, such as accepting and rejecting null hypotheses.

Explained, lines 402-410

Table no. 2 is unreadable.

Corrected with a larger leading (line spacing) value.

In the text informing about the composition of the industry, it would be appropriate to state their full name instead of the abbreviations of the industry. Chapter title 3. "Population", does not reflect the actual contents of the chapter.

A clarification regarding industries has been added to lines 341–351.

The sentence " The unprecedented business consequences include market and financial shock [8], understaffing, and disturbed supply chains [16].“. There indicates that staff shortages were noted during the pandemic. Would you please specify which sector has experienced a need of employees, or the reasons for their lack? According to the development, it is impossible to consider this state of affairs as a dogma; there are known cases where industries such as hotels, restaurants, spas, air services have started redundancies. The car industry has introduced shortened operating modes.

The matter has been clarified and content modified, see lines 105–106.

Reviewer 4 Report

The theme is interesting, however, as mentioned in the text, it is limited to confirming previous works. Lack of innovation. The fact that the sample is very comprehensive (sectors of activity) makes the work very diffuse. I think it could benefit from being more objective and working the results more clearly and succinctly, perhaps choosing one or another sector of activity, or perhaps working only with the size of the companies and testing the 15 hypotheses/variables. Figures 1 and 2 are poorly supported. Since these two models are the truly innovative part of the work, they should be better scientifically based. 

Author Response

Dear Professor,

Thank you very much for all the comments and recommendations in the review. They are invaluable and helped improve the scientific value of the paper. Allow me to clarify certain issues:

The theme is interesting, however, as mentioned in the text, it is limited to confirming previous works. Lack of innovation. The fact that the sample is very comprehensive (sectors of activity) makes the work very diffuse. I think it could benefit from being more objective and working the results more clearly and succinctly, perhaps choosing one or another sector of activity, or perhaps working only with the size of the companies and testing the 15 hypotheses/variables. Figures 1 and 2 are poorly supported. Since these two models are the truly innovative part of the work, they should be better scientifically based. 

The hypotheses have been revised and reduced – lines 254-322. The comment regarding population comprehensiveness is valid , but can hardly be rectified at this stage of the research. The research questions concerned the issue whether all industries experienced the COVID-19 crisis and where it was the most pronounced. Another matter was the question whether the ability to innovate was industry-dependent. The comment will be used in further research. Figures 1 and 2 have been moved to Literature. Information that the models were based on the literature review and original research has been added. See lines 676-688, 691-694, 699, 719-724, and 729.

Round 2

Reviewer 1 Report

  1. The response to the comments and suggestions are too simple, I can not get what and how the paper revised, and some comments in my last round review are not get revised.
  2. this paper has no conclusion, and the section 6, 7 and 8 should be integrated into a whole conclusion section.

Author Response

Dear Professor,
Regarding the comments and suggestions by Reviewer 1, allow me to clarify:
Round 2
1. The response to the comments and suggestions are too simple, I can not get
what and how the paper revised, and some comments in my last round review
are not get revised.
Please excuse my briefness in responses to comments and suggestions, which the
Reviewer believes prevented them from identifying exact improvements to the
manuscript. I strove to implement all corrections pointed out by the Reviewers. To
better present what has been improved and to what degree, please note:
Round 1
1. Through the questionnaire survey, 2000 companies were randomly selected to
conduct the survey by email, but only 622 questionnaire responses were
received. A single survey method results in less data received. Several other
survey methods should be added. Collecting data through multiple channels can
obtain more comprehensive data.
A more detailed description of who was sent the link to the questionnaire has been
added (line 335). Information has been added that the link had also been posted in
social media. Furthermore, please note that it is difficult to add other survey methods
and data acquisition channels at such a late stage of the research. This comment will
undoubtedly be used in future research where I will employ more channels to collect
data. The research was conducted during the COVID-19 pandemic, which made it
difficult to acquire data with interviews, for example.
2. The data source of this article is the questionnaire, and then analyze the
conclusion based on the obtained data. Therefore, the authenticity of the data
is extremely important and is the prerequisite for the analysis to reach the
conclusion. However, whether the data obtained through the questionnaire is
true or not? In order to ensure its authenticity, the author should add an
explanation.
In order to improve the reliability of the research, the questionnaire was aimed at
economically active people (employees, business owners, managers) who are
practitioners and operate in business circles. – line 335-338.
3. The related works about product innovation and managment innovation shoule
be further reviewed, such as A combined fuzzy DEMATEL and TOPSIS
approach for estimating participants in knowledge-intensive crowdsourcing;
Measuring knowledge diffusion efficiency in R&D networks; Selection of
manufacturing enterprise innovation design project based on consumer’s green
preferences; Decision-making and coordination of green closed-loop supply
chain with fairness concern
The content and literature review have been expanded, lines 208–233 and 649–
659.
The following two paragraphs on innovation design based on environmental consumer
preferences and on closed-loop green supply chain focusing on fairness have been
added:
‘They believed the process of innovating alone may be inconsequential for the
economic growth if innovative products are not bought but instead poorly accepted by 
consumers. The lack of demand for new products is a barrier associated with the
implementation of innovation and a ball and chain for the entire innovation process.
The innovative effort of enterprises and the market behaviour of consumers are
significantly linked beyond any doubt. The relationships are investigated on various
levels. K. WÅ‚odarczyk [51] identified key research areas regarding modern consumer
market behaviour associated with introducing innovation to a market: innovative
consumption models, diffusion of innovation, the impact of norms, values, beliefs, and
personality traits on the reception of innovation, new technologies, innovating
consumer behaviour, models of embracing innovation by consumers, and resistance
to innovation from consumer groups.
Analyses of innovative consumption models focus on how consumers use innovative
products and services and what consumers know about sustainable development in
production [52]. Additionally, consumer environmental preferences need to be
considered when designing innovations. Sustainable development of production
makes manufacturers strive towards a competitive advantage through appreciating
public expectations. The increase in respect for the natural environment in business
noticeable for some years has become the primary development trend in the supply
chain, leading to the emergence of green supply chains [53]. One of the founding
fathers who introduced the term into the literature is Beamon [54]. The concept of a
green supply chain involves a comprehensive outlook on relationships between the
natural environment and production optimisation within the supply chain. Green supply
chain management takes into account the entire cycle of product design, production,
packing, sale, use, and recycling, including storage, transport, and information flow
that should conform to environmental standards [55].’
Furthermore, innovation-related processes have been related to knowledge – lines
649–659:
‘Furthermore, the analyses suggest that the introduction of innovation is the
responsibility of managers, leaders oriented towards people and change. Hameed,
Nisar, and Wu [84] discussed the link between leaders and innovations and suggested
that leaders should be oriented towards knowledge. The authors believed that
leadership is among the most potent sources of increased organisational effectiveness
by developing knowledge infrastructure, leading to the strengthening of innovative
solutions. Over the recent years, researchers investigated how management practices
and systems facilitated innovativeness. Some demonstrated that knowledge
management is an important backbone of business innovation [83, 84, 85, 86, 87].
Therefore, leadership-oriented managers will look for knowledge internally and
externally to apply it to new and much-improved products, processes, organisational
structures, and marketing and management.’
The manuscript has not been expanded with the issues of DEMATEL and TOPSIS
approaches to knowledge-intensive crowdsourcing participation estimation or the
effectiveness of knowledge dissipation in R&D networks because the paper is
extensive and focuses on technological and non-technological innovations. The
introduction of such content could make the paper chaotic. The paragraph concerning
future research was expanded with the prospect of interesting investigations into
product innovation and innovation management combined with the fuzzy DEMATEL
and TOPSIS approaches to estimating knowledge-intensive crowdsourcing
participation and the measurement of the efficiency of knowledge transfer in R&D
networks.
4. The author believes that during the COVID-19 pandemic, innovative companies
will have better ways to deal with uncertainty, and therefore companies should 
enhance their innovation capabilities. This result is based on the collected
questionnaires. In order to enhance the credibility of its conclusions, some
examples should be added for comparison, which is clearer and more intuitive.
The paragraph has been expanded with the following piece – line 633–648:
‘The research supported the conclusion that enterprises capable of innovation would
have better ways of handling the uncertainty during the COVID-19 pandemic.
Therefore, the business should improve its innovation capabilities. The result is
consistent with research by experts at the Polish Agency for Enterprise Development
[80], who analysed actions taken globally by various countries to prevent the
consequences of the COVID-19 pandemic crisis. According to their report, an
innovative business approach to the unstable environment was a must during the
pandemic. Experts also confirmed the growing importance of modern technologies,
digitalisation, and sustainable development efforts [80] (orientation towards ecological
preferences of consumers and green supply chains). Experts at the general meeting
of the World Technopolis Association [81] reached similar conclusions and pointed to
innovation as the best way of combating the crisis. Babina, Bernstein, and Mezzanotti
[82] noted that financial crises could act both as destructive and creative forces for
innovation and provided the first systematic evidence of the role of anxiety in the longterm organisation of the innovative business. Moreover, the ability to innovate is one
of the key features of competitive, dynamic, and progressive organisations. [83]’
5. The author proposes and introduces a variety of innovative methods, including
business model innovation, management innovation, etc. However, it is not
enough to introduce the priority of various methods, and you can introduce it in
detail in this aspect.
The models have been described more extensively. Additionally, it has been indicated
that the models were developed based on the literature review and original research.
The following content has been added:
‘Boards and owners of enterprises now face the responsibility of ensuring liquidity and
preserving jobs. It may be the first time some of them came across such a substantial
uncertainty regarding the future. They will have to make strategic-level decisions that
will determine whether and how fast their business will recover from the crisis. Their
success will hinge mostly on the flexible adaptation of the enterprise to market
changes, an achievement for which innovation may be the key [88]. The pandemic can
only kindle innovation: organisations do not grow weaker searching for innovation to
boost effectiveness and optimisation, often technology-based, but embark on the
search for business model innovations with new energy to grow agile and resilient to
the competitors [88, 89], but most of all, to survive. The analyses demonstrated that
innovation is the potion of survival [26, 27, 28, 29, 30, 32, 36], so modern business
models should be based on them. Again, product, process, organisational, marketing
[33, 34, 35] and management [36, 37, 38, 90] innovations should be the primary focus.’
‘As was already mentioned, innovation is a key process within the organisation.
Without new products, modification of production processes, changes in organisation,
marketing, and management, the organisation cannot survive regardless of its
functional profile [91].’
‘It is not enough to determine the business model. Its implementation is also a critical
stage. As shown in the business model, the ability to innovate is primarily based on the 
ability to change combined with creativity. Thus, the four leading components for the
business model are leaders, the ability to change, creativity, and innovation [93, 94,
95, 96].’
Lines 678-690, 693-696, and 721-725 have been expanded.
2. This paper has no conclusion, and the section 6, 7 and 8 should be integrated
into a whole conclusion section.
Sections 6, 7, and 8 have been merged into a conclusions section. Additionally,
sections discussion and results have been split as suggested by Reviewer 2 in Round 1. 

Reviewer 4 Report

Relevant improvements were made and the overall quality of the article improved. 

Author Response

Dear Professor,
In response to the comments of Reviewer 4, I am grateful for the opinion that significant
improvements had been introduced, enhancing the general quality of the paper.

Round 3

Reviewer 1 Report

I have no other comments, it is ok now